# Effects of Selenium Enrichment on Dough Fermentation Characteristics of Baker’s Yeast

**DOI:** 10.3390/foods12122343

**Published:** 2023-06-11

**Authors:** Ping He, Mengmeng Zhang, Yizhe Zhang, Hui Wu, Xiaoyuan Zhang

**Affiliations:** 1College of Food Sciences and Engineering, South China University of Technology, Wushan Road 381, Guangzhou 510640, China; 201910105374@mail.scut.edu.cn (P.H.); zhangmengmengscut@126.com (M.Z.); 202020126409@mail.scut.edu.cn (Y.Z.); 2Industrial Technology Research Institute, South China University of Technology, Guangzhou 510641, China

**Keywords:** *Saccharomyces cerevisiae*, selenium, carbon dioxide, glycometabolism, bread

## Abstract

In this research, the effect of selenium (Se) enrichment on dough fermentation characteristics of yeast and the possible mechanisms was investigated. Then, the Se-enriched yeast was used as starter to make Se-enriched bread, and the difference between Se-enriched bread and common bread was investigated. It was found Se enrichment increased CO_2_ production and sugar consumption rate of *Saccharomyces cerevisiae* (*S. cerevisiae*) in dough fermentation, and had positive impacts on final volume and rheological index of dough. The mechanism is possibly related to higher activity and protein expression of hexokinase (HK), phosphofructokinase (PFK), pyruvate kinase (PK), citrate synthase (CS), isocitrate dehydrogenase (ICD), and α-ketoglutarate dehydrogenase (α-KGDHC) in Se-enriched yeast. Moreover, Se-enriched bread (Se content: 11.29 μg/g) prepared by using Se-enriched yeast as starter exhibited higher overall acceptability on sensory, cell density in stomatal morphology, and better elasticity and cohesiveness on texture properties than common bread, which may be due to effect of higher CO_2_ production on dough quality. These results indicate Se-enriched yeast could be used as both Se-supplements and starter in baked-foods making.

## 1. Introduction

Selenium, an essential trace element for humans, is involved in antioxidant protection and redox regulation as a component of selenoproteins and source of selenometabolites [1]. Se deficiency can affect the immune and reproductive systems and the thyroid function, and is associated with the development of Keshan and Kashin–Beck diseases, among others [2,3]. Se concentration in soil is low in most parts of the world, leading to general dietary deficiency [4]. Therefore, dietary Se supplementation is an important research topic. In nature, Se exists mainly as inorganic Se, but supplementation with inorganic Se is difficult owing to its high toxicity and low bioaccessibility. Microorganisms, especially fungi, are currently the main source of organic Se. Yeast is one of the carriers that can efficiently convert organic selenium, which exhibits a short growth cycle, high tolerance to inorganic Se, and considerable conversion ability [5,6]. At present, Se-enriched yeast has been widely studied and utilized [7].

Yeast is essential in the food industry owing to its superior fermentation characteristics to produce bread, biscuits, beer, wine, and other foods [8]. However, the application properties of Se-enriched yeast in food are currently relatively singular. At present, Se-enriched yeast is mainly used to supplement animal feed to obtain Se-enriched eggs, milk, and other foods [9,10]. Additionally, Se-enriched yeast is often used directly as a Se supplement in the food field. However, the direct addition of Se-enriched yeast may yield low acceptability of food by consumers, because yeast may affect the original flavor, taste, and other sensory qualities of food. In fact, many foods are made using yeast fermentation properties, such as bread and steamed buns. It is speculated that it can considerably increase organic Se content with less adverse impact on flavor and other sensory qualities to make these fermented foods using Se-enriched yeast as fermentation microbes. Such Se-enriched foods may be more easily accepted by the public [11]. Among the various cereal products, bread is still considered as the main part of the diet in most countries of the world [12]. Breadmaking requires yeast to produce large amounts of CO_2_ to produce a loose and porous structure. Additionally, yeast can produce certain flavor substances, such as lactic and acetic acid, ethanol, esters, and other volatile substances, and improve the extensibility and viscoelasticity of dough. Thus, the taste and flavor of bread are therefore closely related to the fermentation characteristics of yeast [13]. However, few studies have focused on the effects of Se enrichment on dough fermentation characteristics of yeast, and the application of Se-enriched yeast as starter in baked foods.

The application of yeast in food is mainly related to the fermentation products obtained from the conversion of sugars under aerobic or anaerobic conditions. The production of alcohol and CO_2_ in yeast fermentation mainly depends on its glycometabolism, which is the main source of carbon elements required by cell synthesis of various substances. These include glycolysis, tricarboxylic acid cycle, trehalose metabolic pathway, Harris pathway, and other metabolic pathways [8,14]. It was reported that selenium can also affect sugar metabolism in animals and plants. The literature indicates that sodium selenate can significantly increase the activity of enzymes related to glycolysis in potatoes, such as pyruvate kinase and phosphofructose kinase [15]. Similarly, selenium has been reported to affect sugar metabolism in animals by affecting insulin synthesis and the body’s sensitivity to insulin [16]. In fact, in addition to animals and plants, some studies have reported that selenium enrichment leads to high levels of pyruvate kinase, glucokinase, triose phosphate isomerase, and other related enzymes in glycometabolism pathways in baker’s yeast through proteomics. These studies indicate that selenium can affect the glycometabolism of living organisms, but there is currently no research exploring the effect and possible mechanism of selenium enrichment on yeast glycometabolism pathways. Therefore, exploring the impact and mechanism of selenium enrichment on yeast glycometabolism can provide a scientific basis and important theoretical significance for studying the fermentation characteristics of Se-enriched yeast and expanding its application scope.

In summary, most of the current studies on Se-enriched yeast focus on the form of Se, the preparation of Se-enriched yeast, and the effects of Se-enriched yeast as a source of Se supplement on animal physiological health. There are few studies on the effects of Se enrichment on fermentation characteristics and glycometabolism of yeast. In response to the current research situation, the effect of Se enrichment on dough fermentation characteristics of yeast was investigated and explored in this study, including CO_2_ production, reducing sugar consumption, and other indicators. Furthermore, Western blot was used to explore the laws and possible mechanisms of Se enrichment affecting yeast glycometabolism. Then, the Se-enriched yeast was used as starter to make Se-enriched bread, and the difference of sensory quality, stomatal morphology, textural properties, and Se content between Se-enriched bread and common bread was investigated. On the one hand, the results of this study establish a foundation for the application of Se-enriched yeast as a Se supplement and starter in the production of Se-enriched fermented foods, and provide a new direction for the application of Se-enriched yeast. On the other hand, it deepens the understanding of the effect of Se enrichment on glycometabolism in yeast and the theoretical system of the mechanism of Se enrichment on glycometabolism.

## 2. Materials and Methods

### 2.1. Materials and Chemicals

*Saccharomyces cerevisiae* (baker’s yeast, GDMCC 2.167), often called baker’s yeast, was purchased from Guangdong Microbiological Species Preservation Center, China. Yeast was grown in yeast extract–peptone–dextrose (YPD) medium (comprising 1% yeast extract, 2% glucose, and 2% peptone) obtained from Qinghai Haibo Biological Co., Ltd., China. Solid medium was obtained via the addition of 2% agar [7]. Sodium selenite, α-amylase (porcine pancreas, ≥5 U/mg solid), and pepsin (porcine gastric mucosa, 3000 U/mg) were purchased from Sigma-Aldrich Corporation, St. Louis, MO, USA. Trypsin (bovine pancreas, 8 U/mg) was obtained from China Yuanye Co., Shanghai, China. Distilled water was supplied by a Millipore Milli-Q water purification system. HK, PFK, PK, CS, ICD, and α-KGDHC kits were purchased from Shanghai Absin Biotechnology Co., Shanghai, China. All antibodies for Western blotting were obtained from Cell Signaling Technology (Beverly, MA, USA). Other analytical grade chemicals and reagents were purchased from local commercial sources.

### 2.2. Preparation of Se-Enriched Yeast

*Saccharomyces cerevisiae* strains used in baking were cultivated to produce fresh Se-enriched yeasts. Activated *S. cerevisiae* was inoculated at 10% (*v*/*v*) in 100 mL YPD liquid medium. An appropriate amount of sodium selenite mother liquor was added to produce a final concentration of 30 µg/mL. Cells were grown at 28 °C and 180 r/min for 72 h. The resulting culture was centrifuged at 5000 r/min for 10 min, washed three times with distilled water to remove adsorbed Se, and collected and preserved as Se-rich yeast [17].

### 2.3. Determination of Se Content and Speciation

The determination of Se content was slightly modified from the method of Liu et al. [9]. Five-tenths gram samples were digested with aqua regia for wet digestion, and total Se content was measured using hydride generation atomic fluorescence spectrometry. An additional 0.5 g was placed in a test tube. HCl (20 mL, 50%) was added, and the mixture sonicated at 25 °C for 45 min. The sonicated sample was placed in a boiling water bath for 30 min and then cooled. Filtrate was collected by suction using a Buchner funnel. Then, 5 mL of filtrate was placed in a small beaker, and aqua regia was used for wet digestion. Total inorganic Se content was measured by hydride-generation atomic fluorescence spectrometry. The difference between total and inorganic Se estimates organic content.

Speciation of Se is based on reports by Moreda-Piñeiro et al. [18] and slightly modified. Selenomethionine (SeMet), selenocysteine (SeCys), and methylselenocysteine content and speciation were determined using high-performance liquid chromatography–inductively coupled mass spectrometry (HPLC–ICP-MS). Waters AccQ.Fluor reagent was used for derivatization in an AccQ.Tag column (Waters Nova-Pak C184 mm, column 3.9 ± 150 mm). A Waters HPLC system with a 1525 binary pump and multifluorescence detector (Waters 2475) was employed.

### 2.4. Dough Fermentation by Se-Enriched Yeast

Briefly, 5 g white granulated sugar and 1.0 g NaCl were mixed in 55 mL purified water. Next, 100 g refined wheat flour (14% water, 10.7% protein, and 0.4% ash), 2 g of yeast or Se-enriched yeast, and 5 g butter were added for hydration. The dough was first mixed in an 80 r/min vertical mixer for 10 min until even and smooth. The dough was then transferred to a fermentation box at a temperature of 30 °C ± 0.5 °C and 75% relative humidity for 3 h. After fermentation, the dough was cut into equal-weight portions and placed in a fermentation box at 30 °C ± 0.5 °C and 85% relative humidity for 30 min until the dough was shaped.

### 2.5. Determination of Reducing Sugar Content

The reducing sugar content in dough during fermentation was determined using HPLC [19]. Dough samples fermented at different time periods were weighed and mixed with absolute ethanol at a ratio of 1:3 (*m*/*v*), placed in a water bath at 70 °C for 30 min, and centrifuged at 3000 r/min for 10 min. Then, the supernatant (1 mL) after centrifugation as described above was taken, mixed with 3 mL absolute ethanol again, and centrifuged at 10,000 r/min for 15 min to retain the supernatant. Finally, the supernatant was filtered through a 0.22 μm membrane for HPLC analysis. The chromatographic column was a Shodex Asahipak NH2 P-50 4E (4.6 mm × 250 mm). The column temperature was 30 °C, the detector was set for evaporative luminescence, the mobile phase was acetonitrile: H_2_O (*v*/*v*) = 3:1, and the flow rate was 1.0 mL/min. The injection volume was 10 μL. An external standard (fructose, glucose, and maltose) was used for quantification.

### 2.6. Determination of CO_2_ Production and Dough Volume

Shaped dough was immediately placed into the stainless steel box of a fermentation tester (WSF-2000MH, Tokyo, Japan) and sent to the vitality room. Fermentation temperature was set at 30 °C ± 0.5 °C. The tester was adjusted, and the CO_2_ gas production (mL) was recorded over time via water yield. The dough was placed in the sample room of a food volume analyzer (JMJY, Chengdu, China), and the volume occupied by the dough squeezed rapeseed oil in the sample room into a graduated glass tube. Then, the volume of the dough (mL) was recorded according to the scale.

### 2.7. Determination of Dough Rheological Fermentation Characteristics

The test simulated dough fermentation processes over 5 h and was tested every 30 min. Shaped dough with an initial weight of 315 g was used to assess the characteristics of dough fermentation with a rheological fermentation apparatus (F3 Rheofermentometer, Chopin Technology Co., Paris, France). The test conditions were: experimental temperature of 30 °C ± 0.5 °C, counterweight weight of 2 kg, standard piston, and test period of 3 h [20].

### 2.8. Key Enzyme Activities in Se-Enriched Yeast

After culturing, yeast was transferred to a normal YPD medium. After 24 h, fermentation broth was centrifuged at 3000 r/min for 15 min. Pellets were washed with sterile normal saline three times, then suspended in 0.02 M phosphate buffer of pH 7.0. Ultrasonication to crush the pellets was repeated 20 times at intervals of 2 s and 10 s, and the suspensions were then centrifuged at 10,000 r/min for 10 min to remove cell fragments. The supernatant was used for enzyme assays.

Enzyme contents were assessed using appropriate test kits, following the manufacturer’s instructions. The determination principle was as follows: purified human HK, PFK, PK, CS, ICD, and antibody against α-KGDHC were coated on microplates to prepare solid-phase antibody. Enzymes were added to wells coated with monoclonal antibodies, and then combined with horseradish peroxidase (HRP)-labeled enzyme antibodies to form antibody–antigen enzyme-labeled antibody mixtures. After washing, 3,3′,5,5′-tetramethylbenzidine substrate was added for color development, which was converted to blue under HRP catalysis and finally to yellow at low pH. The depth of color is positively correlated with enzyme activity. Absorbance value was recorded with a microplate reader at 450 nm and enzyme concentrations calculated using sample solution standard curves [21].

Protein expression of HK, PFK, PK, CS, ICD, and α-KGDHC in yeast cells were analyzed using Western blotting. Quantitative analysis used ImageJ with β-actin as an internal reference, and relative expression level of protein = expression of target protein/expression of internal reference protein.

### 2.9. Bread Preparation

The optimized sponge breadmaking method, AACC 10-11.01 (AACC., 2000), was adopted with a slightly modified formula [22]. These breads were produced for sensory and chemical analysis. The final dough was baked at a constant temperature (180 °C on upper heat and 200 °C on lower heat) for 25 min. The bread was then cooled to room temperature and stored sealed in polyethylene bags at room temperature for 48 h for index analysis.

### 2.10. Determination of Se Bioaccessibility in Bread

Determination of Se bioaccessibility is based on reports [18,23], with minor modifications. Briefly, 1.00 g of bread crumbs was mixed and shaken with 5.0 mL water and 5 mg α-amylase. This mixture was incubated at 37 °C for 15 min. The sample solution was diluted to 10 mg/mL with water until 100 mL and stirred until uniform. The pH of the sample solution was adjusted to 3.0 using 1 M HCl. Pepsin (66.7 mg, 3000 U/mg enzyme activity) was added and stirred magnetically at 37 °C for 2 h. Subsequently, the pH of the sample was adjusted to 7.0 using 1 M NaOH to inactivate pepsin and 1250 mg pancreatin (8 U/mg) was added. Samples were stirred at low speed for 2 h at 37 °C. Finally, the pH of the digestive solution was adjusted to 7.0 and placed in a boiling water bath for 10 min to inactivate pancreatin and trypsin. Digested samples were sonicated for 30 min to fully release selenoaminoacids, and then centrifuged at 4 °C and 8000 r/min for 10 min. The supernatant—the in vitro simulated gastrointestinal digestive product—was filtered through a sterile 0.45 mm filter (Corning, NY, USA) and prepared for analysis. The subsequent analysis method was consistent with the Se speciation analysis described in Section 2.3.

### 2.11. Evaluation of Sensory, Stomatal Morphology, and Textural Properties in Bread

Sensory assessment was performed by 40 trained panel members. Bread samples were stored at room temperature for 4 h. Each team member simultaneously received a test sample (three copies of each sample), glass of water, and ballot, and performed sensory evaluation on the appearance, color, flavor, taste, and overall acceptability using a 9-point hedonic scale [11,24]. All sensory tests were performed in a sensory assessment laboratory. In addition, the morphology structure of bread cross-sections was evaluated using HD camera. Bread slices (8.5 cm × 8.5 cm × 1.5 cm) were selected and placed on the test platform. Images were captured using an HD camera (Canon, Japan, 1x). The analysis of stomatal morphology in cross-sections of bread is based on a previous study [25], with a slight modification. The images were processed using ImageJ version 1.50i (National Institute of Health, Rockville, MD, USA). The image resolution was set at 300 dpi. The visual field of 2 cm × 3 cm in the bread center was captured and selected. Cell density (Number of stomata/Field area, cell/cm^2^), average stomatal area (Average stomatal area/Number, mm^2^), and area fraction (Stomatal area/Field area, %) were calculated.

Then, bread slices (8.5 cm × 8.5 cm × 1.5 cm) were placed on the test platform using the texture profile analysis mode as described by Guardado-Félix et al. [24], with partial modification. The specific parameters were: pretest speed of 1.00 mm/s, test speed of 1.00 mm/s, and post-test speed of 2.00 mm/s. Compression ratio was 50% and load at the trigger point was 5 g. The interval between compressions was 3 s. The lowest and highest values of 10 parallel measurements were discarded and an average was calculated. Five physical properties were recorded from the texture test curve, including hardness, elasticity, cohesiveness, chewiness, and resilience.

### 2.12. Statistical Analysis

One-way analysis of variance was used to analyze differences between the groups by using SPSS 21.0. Results were considered statistically significant when *p*-value < 0.05. Data are expressed as mean ± standard deviation, n = 3. 

## 3. Results and Discussion

### 3.1. Content and Speciation of Se in Se-Enriched Yeast

*Saccharomyces cerevisiae* (baker’s yeast) is commonly used in dough fermentation and then making baked food. The laboratory-produced Se-enriched yeast also shows a substantial ability to accumulate Se [26]. Thus, this strain was selected for dough fermentation and further baking food production. In addition, under the same conditions, initial cultivation was carried out using Se-enriched yeast and original yeast of the same weight; there was no significant difference in the total number of colonies formed by the two strains in final, *p* > 0.05. The content and speciation of Se in Se-enriched yeast are summarized in Appendix A. The total Se content in laboratory-produced Se-enriched yeast was 3150.56 μg/g, much higher than that found in commercial Se-enriched yeast. The content in commercial Se-enriched yeast is typically 500~2000 μg/g. Commercial yeast is generally *S. cerevisiae* discarded by the beer industry and processed for Se enrichment [27]. It has been reported that the enrichment site of Se in yeast is mainly concentrated in the sediment from protoplast fragmentation, and the Se content of this part accounts for more than 75% of the total Se content [27]. However, brewing materials, alcohol, and other metabolites often form high osmotic pressure in the process of beer production, which to some extent causes the plasmolysis in yeast. This leads to damage in the protoplasts of yeast, and reducing the ability of waste yeast from the beer industry to accumulate Se. The yeast selected herein is a fresh, robust strain with substantial ability to accumulate Se. Therefore, the Se enrichment ability of *S. cerevisiae* (GDMCC 2.167) used in this study is stronger than that of commercial yeast.

Moreover, speciation indicates mainly SeMet (2376.89 μg/g), which accounts for 75.44% of total Se content. Due to the similarity in chemical properties between Se and sulfur, inorganic Se is often transformed into organic Se through the sulfur metabolism pathway of organisms and performs its biological functions. There are differences in the metabolic pathways of seleno-amino acids in different organisms. The cysteine synthase pathway of yeast is weak, and it mainly realizes the large amount of SeMet accumulation through the homocysteine synthase pathway. In addition to this, SeMet itself is more stable than SeCys. SeCys is easily oxidized to methyl- SeCys, while methyl- SeCys cannot participate in protein synthesis. When it cannot be used as an amino acid raw material to participate in protein synthesis, it cannot accumulate in large quantities in cells, and it is quickly metabolized. However, SeMet can participate in protein synthesis by replacing the form of methionine, thereby achieving a large accumulation in proteins [7]. Therefore, this is why Se-enriched yeast is mainly in the form of SeMet. Similarly, Sanchez-Martinez et al. [28] reported that the Se species in Se-enriched yeast is mainly SeMet (70.88%), close to the value herein. There are also peaks of other Se species in Se-enriched yeast, which may indicate Se polysaccharide and a small amount of SeCys [29].

### 3.2. Effects of Se-Enriched Yeast on the Reducing Sugar Consumption, CO_2_ Production, and Dough Volume of during Dough Fermentation

The main role of yeast in dough fermentation is to produce CO_2_ gas through consumption of reducing sugar; gas is retained in the dough by a network of tissue formed by gluten, giving the dough soft and porous characteristics, thus bringing good sensory quality to the bread. In order to investigate the effect of Se enrichment on the dough fermentation characteristics of yeast, the reducing sugar consumption rate and CO_2_ production of yeast during dough fermentation were measured. The changes in reducing sugar content after 24 h of fermentation are shown in Figure 1A. Se-enriched and original yeast exhibit different consumption patterns. Se-enriched yeast induced a rapid decrease in reducing sugar content over the first 4 h and stabilized after 4–24 h. However, the reducing sugar content of dough fermented by the original yeast strain continued to decrease over the initial 8 h and stabilized after 8–20 h. Se-enriched yeast consumes reducing sugars faster than the original yeast. In the breadmaking process, fermented dough generally lasts for three 6 h cycles. During cycling, Se-enriched yeast consumes reducing sugars relatively rapidly. Additionally, significant differences were observed in CO_2_ release during the fermentation process (*p* < 0.05) (Figure 1B). The two kinds of yeast exhibited healthy aerobic respiration at the start of fermentation, and fermentation was relatively fast. After some time, CO_2_ gas production increased slowly to a flat level. Se-enriched yeast reached the maximum gas-released volume of 1454.5 mL at a relatively fast speed in the initial fermentation stage. After 3 h, CO_2_ gas production was stable. In contrast, the original yeast fermented relatively slowly, and gas-released volume gradually increased to a maximum of 1325.5 mL. After 4 h, CO_2_ gas production gradually stabilized. Overall, fermentation by Se-enriched yeast was relatively rapid compared to that with the original yeast strain, and the final CO_2_ gas production was significantly higher (*p* < 0.05). Studies have shown that Se enrichment can enhance the rate of reducing sugar consumption in *Candida utilis* [30]. The research results of the transcriptome showed that Se enrichment can significantly affect the expression of glycometabolism-related enzymes in *Candida utilis* [31]. Ekumah et al. [32] reported that low concentrations of Se could promote the rate of CO_2_ release during the fermentation of mulberry juice by *S. cerevisiae*, ultimately increasing the alcohol production. In fact, in addition to yeast, Se also affects glycometabolism in plants and animals. For example, literature reports have shown that sodium selenate can significantly improve the activity of enzymes related to glycolysis such as PK and PFK during potato growth [15]. Malik et al. [33] reported that the increase in mungbean growth of shoots and roots by application of Se was the result of upregulation of enzymes (sucrose synthase and sucrose phosphate synthase) in glycometabolism, thus providing energy substrates for enhanced growth. Moreover, Se enhances the mRNA and protein expression levels of key proteins in the insulin signaling pathway (PI3K-Akt-GSK3) in rat liver cells, including glucokinase (GK), glycogen synthase (GS), and glycogen synthase kinase 3(GSK-3), thereby promoting the upregulation of glycometabolism levels in rat liver cells [34]. From this, it can be seen that exogenous Se addition may affect the cellular glycometabolism pathway, thereby altering the sugar consumption rate and CO_2_ production of cells.

Dough volume also increased with increasing fermentation time and then decreased to a stable volume (Figure 1C). Dough contains a certain amount of air early in the fermentation process; yeast initially uses this air for substantial aerobic respiration, and then switches to anaerobic fermentation with decreased gas production. The continuous production of gas causes air holes in the dough to break and allow gas to escape. At such time, the height of the dough will not increase even as gas generation continues. During the first 2 h of fermentation (0–2 h), the dough volume supported by Se-enriched yeast increased relatively quickly. At 1.5 h, the dough volume was close to maximum. In contrast, growth in dough volume by the original yeast strain was slower and reached a maximum volume after fermentation for 2.5 h. After 3 h, the dough volume for both types of yeast was stable at its highest value. The final volume of dough produced by Se-enriched yeast (350.2 mL) was significantly higher than the volume of dough produced by the original yeast (335.7 mL) (*p* < 0.05). This finding is consistent with the actual trend in CO_2_ gas production discussed above in Figure 1B. Xie et al. reported that the glucose repression effect on yeast was removed by controlling glucose levels during fermentation, resulting in an increase in the fermented dough volume, which was attributed to the accelerated utilization of reducing sugar by yeast and increased carbon dioxide production during dough fermentation. Therefore, in this study, the increase in dough volume may also be related to the accelerated utilization of reducing sugar by Se-enriched yeast during dough fermentation, which leads to the increase in carbon dioxide [21].

### 3.3. Effects of Se-Enriched Yeast on the Rheological Fermentation Characteristics during Dough Fermentation

Rheological fermentation characteristics by Se-enriched yeast during dough fermentation were further quantified using an F3 fermentation rheometer. The maximum expansion of dough and total volumes of gas directly reflect yeast gas production. Maximum swelling height (H_m_) and total volume of the released gas (V_gas_) by Se-enriched yeast are higher (44.90 ± 0.35 mm and 1687.53 ± 53.75 mL, respectively) than those for ordinary yeast (Table 1), reflecting greater final dough volume. This finding is consistent with the previous CO_2_ production measurements. The time when the Se-enriched yeast fermented dough began to leak CO_2_ (T_X_ = 79.50 ± 0.60 min) was earlier than that for the original yeast strain (85.50 ± 0.35 min) because the rate of gas production in the early stage of fermentation was faster (Figure 1B). Good gas production reflects relatively high dough leavening power to some extent, which directly affects the final bread quality. In general, Se-enriched yeast exhibits superiority in gas production capability. Furthermore, the gas retention capacity during dough fermentation by Se-enriched yeast (87.15 ± 0.22%) was higher than that for the original yeast strain (83.96 ± 0.25%) (*p* < 0.05). It is reported that yeast plays a certain role in increasing gluten extension in dough fermentation [25]. Whenever CO_2_ is released by yeast metabolism, these gases will disturb surrounding molecules, which will increase the chance for protein molecules to contact with water molecules, so that glutenin in flour will be properly changed. In fact, this effect can be seen as a kind of “kneading” at the molecular level, promoting the formation of gluten and creates conditions for maximum expansion during the awakening and baking stages. This has an impact on the rheological fermentation characteristics of dough, especially in terms of the gas retention (R) and maximum swelling height (Hm) of dough. Se-enriched yeast can not only accelerate the rate of CO_2_ production, but also better retain the gas generated during fermentation in the dough, which may be attributed to the better gluten structure formed by Se-enriched yeast during dough fermentation. Similarly, Se has been reported to confer better dough rheology, bread features, and nutritional properties on yeast-fermented chickpea breads. In particular, the gluten expansibility in dough fermentation was significantly better than that in the control group [11].

### 3.4. Effect of Se Enrichment on Activities of Key Enzymes in Glycometabolism Pathways of Yeast

In the results discussed above, Se-enriched yeast showed faster CO_2_ generation and sugar utilization rates. Sugar consumption and CO_2_ production by microorganisms depend on glycometabolism, such as the Embden–Meyerhof–Parnas (EMP) and tricarboxylic acid (TCA) pathways. Therefore, the activity of key enzymes in the glycometabolism pathway was further determined to explore the relevant impact mechanism of Se enrichment on glycometabolism of yeast. Specifically, reasons for differences in the fermentation characteristics between yeast strains were assessed by examining HK, PFK, and PK enzymes in glycolysis (Figure 2A). This process is critical for glycometabolism in mammals, plants, and fungi. HK catalyzes the first step of glycolysis. PFK catalyzes the reaction of fructose 6-phosphate to produce fructose 1,6-diphosphate, and PK catalyzes the transfer of phosphate phosphoenolpyruvate to ADP. The activities of these enzymes in Se-enriched yeast are higher than those for the original yeast strain. The differences in HK and PK are particularly obvious. Thus, Se-enriched yeast exhibits upregulation of these enzymes, thereby promoting the EMP pathway to enhance glycometabolism. In the actual breadmaking process, yeast basically ferments in an anaerobic environment, which is metabolized through glycolysis (EMP pathway), converting glucose into CO_2_ and ethanol, and producing energy. Yeast after Se enrichment (Se-enriched yeast) promoted the activity of three rate-limiting enzymes (HK, PFK, and PK) in EMP pathway to a certain extent, and finally led to the acceleration of CO_2_ production during the fermentation of Se-enriched yeast. This is consistent with the data trend of fermentation characteristics of Se-enriched yeast that was actually measured. Similar results have been reported elsewhere in the literature. Se supplementation can enhance the mRNA and protein levels of HK, PK, and PFK in chicken spleen, thereby increasing energy metabolism in cells [35]. Moreover, the activities of the glycolytic marker enzymes HK, PFK, and PK were increased 1.7-fold to 3-fold in liver and/or adipose tissue by selenate treatment as compared to mice on Se deficiency [36].

Furthermore, the TCA cycle, respiratory chain reactions, and ATP synthesis are three primary biochemical pathways in mitochondria. The bottleneck in glycometabolism in *S. cerevisiae* is the TCA cycle. CS, ICD, and α-KGDHC are recognized as rate-limiting enzymes that catalyze irreversible reactions in this cycle. The specific activities of these enzymes in original yeast were significantly lower than those in Se-enriched yeast (*p* < 0.05) (Figure 2B). The change in ICD activity was most obvious. The TCA cycle is essential for energy metabolism. Previous reports found that Se-enriched *S. cerevisiae* can promote cell growth and consumption of nutrients, especially in the early stage of culturing [37,38]. We can speculate that Se-enriched yeast may promote intracellular ATP accumulation, achieved through glycometabolism, including enzymes in the TCA cycle. In the early stage of dough fermentation, the system was under aerobic conditions, and yeast metabolized glucose to produce CO_2_ and H_2_O mainly through aerobic respiration (glycolysis → TCA cycle). Se-enriched yeast exhibited higher activity for three key enzymes in the TCA cycle compared to original yeast, promoting the TCA cycle to some extent, encouraging rapid budding and propagation, and ensuring subsequent improved fermentation and CO_2_ generation. Atef et al. [39] reported that Se supplementation increased the activity of CS in ovarian tissue of PCOS model rats. Sinha et al. [40] collected serum samples from thirty-six healthy African American and Caucasian men, indicating that Se supplementation upregulated the protein expression of CS, ICD, and α-KGDHC in healthy male serum.

The expression of six key enzymes in glycometabolism by Se-enriched and original yeasts was quantitatively determined using Western blotting (Figure 2C,D). The original and Se-enriched yeast strains exhibit large differences in protein band expression. Protein expression of HK, PK, PFK, CS, and α-KGDHC in the Se-enriched strain increased, and ICD expression did not change. Upregulation of HK and PFK is particularly obvious. Se enrichment directly affects the expression of HK, PK, PFK, CS, and α-KGDHC. Notably, the expression of ICD protein and activity exhibit a reverse trend. It has been reported that Se supplementation can promote post-translational modification of proteins related to growth and development in yak oocytes [41]. Therefore, this difference may be due to Se enrichment that also affects protein maturation after translation in yeast.

The specific activities of six key enzymes (HK, PFK, PK, α-KGDHC, CS, and ICD) in the Se-enriched yeast were higher than those in the original yeast strain. It can be seen that Se enrichment plays a positive role in regulating the specific activity of this key enzyme in glycometabolism of yeast. Finally, according to these conclusions, it can be inferred that Se enrichment affects the whole glycometabolism pathway of yeast, as shown in Figure 2E, and the specific mechanisms of influence need further research and discussion. Glycometabolism is the main source of carbon element needed by cell synthesis of various substances. There are reports in the literature indicating that during the Se enrichment process in yeast, carbon mainly flows toward amino acid metabolism [31]. In addition, Se supplementation can significantly improve the selenoprotein containing Se-Cys in mammalian cells, and the synthesis of Se-Cys depends on serine. It has also been reported that Se supplementation may affect glycometabolism in mammalian cells through de novo synthesis of serine [42]. These studies may suggest that the effect of Se enrichment on yeast glycometabolism may be related to the metabolism of amino acids. Furthermore, 60–85% of Se speciation in Se-enriched yeast is SeMet [7], which means that Se-enriched yeast needs to synthesize a large amount of SeMet in the process of converting inorganic Se into organic Se. The biosynthesis pathway of SeMet is the same as that of Met. It has been reported that there is interaction between the Met synthesis pathway and glycometabolism in yeast cells [43]. However, to explore whether the effect of Se enrichment on yeast glycometabolism is related to the synthesis of SeMet needs further study.

### 3.5. Content, Speciation, and Bioaccessibility of Se in Se-Enriched Bread

Total Se (Table 2) in processed Se-enriched bread by Se-enriched yeast as starter was 11.29 μg/g, and most Se was in the form of SeMet (9.24 μg/g). Total Se in common bread was 0.14 μg/g, which could be attributed to the Se in flour and other raw materials. The Se content of Se-enriched bread processed by laboratory-made Se-enriched yeast was significantly higher (*p* < 0.05). Thus, Se-enriched yeast can significantly increase the content of Se in bread. In addition, the proportion of SeMet in Se-enriched yeast itself is 75%. After fermentation and processing into bread, the proportion of SeMet is 81%. Dong et al. [44] reported that Se-enriched potatoes undergo a certain degree of transformation in Se speciation after subsequent processing, such as boiling in water and frying in oil. This suggests that the method of food processing may cause changes in the Se form. Therefore, the increase may be due to the conversion of other forms of Se into SeMet during dough fermentation or subsequent high-temperature baking.

Many Se-enriched foods have been prepared as part of the current research. However, the low bioaccessibility of Se in these foods limits their practical application as a dietary Se supplement. The bioaccessibility of Se in Se-enriched bread as a supplement was investigated using in vitro gastrointestinal simulation digestion (Table 2). The bioaccessibility of total Se after simulated digestion was 94.15%. Vu et al. [45] reported the bioaccessibility of Se in Se-chlorella (49%) and commercial Se-supplements (32%). Thus, the bioaccessibility of Se in Se-enriched bread herein was significantly higher than that in these Se supplements. Research shows that the main forms of organic Se (biological Se) are selenoprotein and Se-polysaccharides, among others [44]. However, some types of Se-polysaccharides cannot be digested and absorbed by the human body. At present, selenoprotein is the main form of dietary Se supplement with the highest safety and absorption efficiency. Specifically, selenoprotein is mainly composed of Se-containing amino acids in the form of SeMet and Se-Cys. Khanam et al. [46] reported the bioaccessibility of Se in some common Se-enriched agricultural products, including Se-enriched cereals (10–24%), Se-enriched legumes (12–29%), and Se-enriched leafy greens (10–31%), and further suggested that the bioaccessibility of Se was mainly attributed to the presence of SeMet. Due to the stability of SeMet, most of it is retained after gastrointestinal digestion and finally utilized by the human body. However, Se-Cys is unstable and easily degraded after gastrointestinal digestion to remove Se-containing groups. Similarly, Thiry et al. [47] reported that the bioaccessibility of Se in Se-enriched yeast was 86.22%, and the proportion of SeMet may be crucial in the digestion and absorption of Se in yeast, as it affects bioaccessibility and bioavailability. Moreover, compared with other Se-enriched foods, the main form of Se amino acids in Se-enriched yeast is SeMet, which has certain advantages in Se bioaccessibility. Therefore, the high bioaccessibility of Se in Se-enriched bread may be attributed to SeMet as the primary Se species. SeMet is the primary species in the bioaccessible fraction in Se-enriched bread. SeMet was not detected in common bread after simulated gastrointestinal digestion in vitro, but the in vitro bioaccessibility of SeMet in Se-enriched bread was 97.51%. Se-enriched yeast remains in bread, which results in the retention of SeMet with associated high bioaccessibility. Se-enriched bread consumption was estimated using information from the Healthy Nutrition Guideline. A piece of Se-enriched bread weighing 5–7 g will provide 50–70 μg Se, which meets the recommended daily allowance range of 55–70 μg/day. Hence, Se-enriched bread can be a suitable source for a dietary Se supplement.

### 3.6. Effects of Se-Enriched Yeast on Sensory, Stomatal Morphology, and Textural Properties of Bread

In order to understand the difference in sensory quality between Se-enriched bread and common bread, 40 team members performed a sensory evaluation. It can be seen from Figure 3A that both bread fermented by Se-enriched yeast and common bread are acceptable, with overall acceptability scores of >7.0. However, Se-enriched yeast has a positive impact on sensory quality, and the overall acceptability reaches 8.6. Bread fermented with Se-enriched yeast was superior to common bread in appearance, taste, and flavor. Particularly, the taste of bread fermented by Se-enriched yeast was significantly better than that for common bread (*p* < 0.05). The addition of Se-enriched yeast improves the texture of bread, increasing the overall sensory score of Se-enriched bread. Moreover, Figure 3B–E show the scanned images of the Se-enriched bread and common bread cross-section and the corresponding images processed by Image J software, which can be used to intuitively analyze the stomatal distribution and morphology. Table 3 exhibits the specific parameters of stomatal distribution in the bread, including cell density, average stomatal area, and area fraction. The distribution of stomata in bread is an important index to reflect the textural structure of bread, which can directly reflect the characteristics of yeast in dough fermentation, and is related to the sensory quality and texture characteristics of bread. As can be seen from Figure 3B–E and Table 3, the cell density and area fraction in the cross-section in bread fermented with Se-enriched yeast were significantly increased (*p* < 0.05), indicating that there were more pores in the cross-section of the Se-enriched bread compared with the common bread. According to the report, more pores in the bread’s cross-section indicate better sensory quality and textural characteristics [11]. In addition, it is reported that the addition of exogenous amylase can ultimately improve the sensory quality and give better stomatal morphology distribution in bread, which is attributed to a significant increase in CO_2_ production during the initial stage of dough fermentation [48]. Therefore, the increase in CO_2_ production by Se-enriched yeast during dough fermentation can result in better cell density and area fraction in bread, thereby achieving good sensory quality. However, compared to common bread, the surface of the bread fermented by Se-enriched yeast had a small number of irregular pores (Figure 3B,C), and the average stomatal area of the cross-section in Se-enriched bread (0.32 ± 0.0012 mm^2^) is larger than that in the common bread (0.24 ± 0.0021 mm^2^), further illustrating the problem. Analyzing the reason, this may result from the relatively fast CO_2_ production rate during dough fermentation by Se-enriched yeast, so that the average stomatal area in bread slightly increases. By fermenting chickpea flour into dough using Se-enriched yeast, Lazo-Vélez et al. [11] reported that the cell density in Se-enriched chickpea bread was significantly higher than that for the control group, which is consistent with the results of this study.

The texture of bread is closely related to its quality. Five indices of hardness, elasticity, cohesiveness, chewiness, and resilience were used to further evaluate the textural differences between Se-enriched and common bread (Table 3). The hardness (113.91 ± 3.33 g) and chewiness (102.68 ± 4.32 mJ) of bread fermented with Se-enriched yeast were significantly lower than those of bread fermented with an original yeast strain (192.67 ± 3.49 g and 114.75 ± 2.19 mJ, respectively) (*p* < 0.05), while its elasticity (1.36 ± 0.02 mm) and cohesiveness (0.90 ± 0.0056 mJ) were significantly higher (*p* < 0.05). However, the difference in resilience was not significant (*p* > 0.05). It has been reported that there is a significant negative correlation between cell density and bread hardness, and a significant positive correlation between pore density and bread elasticity [28]. Due to the increased CO_2_ production and rate during the fermentation process by Se-enriched yeast, together with better gas retention ability, Se-enriched bread has a better value for cell density, which endows it with better elasticity and to some extent reduces the hardness. Thus, bread was endowed with better textural properties. Overall, Se-enriched bread is superior to common bread in terms of sensory quality and textural properties. Moreover, other metabolisms of yeast during dough fermentation also affect the sensory quality of bread. Yeast produces alcohol during dough fermentation, which is then metabolized into esters, organic acids, aromatic alcohols, and other aromatic substances, which together form pleasant aromas and flavors after baking. According to reports, mulberry wine fermented with Se-enriched yeast had higher content of total polyphenols, total flavonoids, total anthocyanins, and types of flavor substances compared to the control group [32]. The quality evaluation model constructed through principal component analysis combined with sensory evaluation showed that the mulberry wine fermented with Se-rich yeast had the best quality. Therefore, the flavor of dough products fermented by Se-enriched yeast may also be improved, and the utilization of carbon sources by Se-enriched yeast is significantly enhanced, thus upregulating the metabolic pathway for the production of flavor substances.

## 4. Conclusions

In conclusion, Se enrichment can improve the specific activity and expression of HK, PFK, PK, α-KGDHC, CS, and ICD enzymes in yeast, thereby increasing reducing sugar consumption rate and CO_2_ production of yeast in dough fermentation, and improving the properties in rheological fermentation of dough, and further improving the sensory, stomatal morphology, and textural properties of bread. Therefore, Se-enriched yeast can be used in bread production as starter and dietary Se supplement without lowering the sensory quality of bread, and can provide an acceptable Se-enriched food.

## Figures and Tables

**Figure 1 foods-12-02343-f001:**
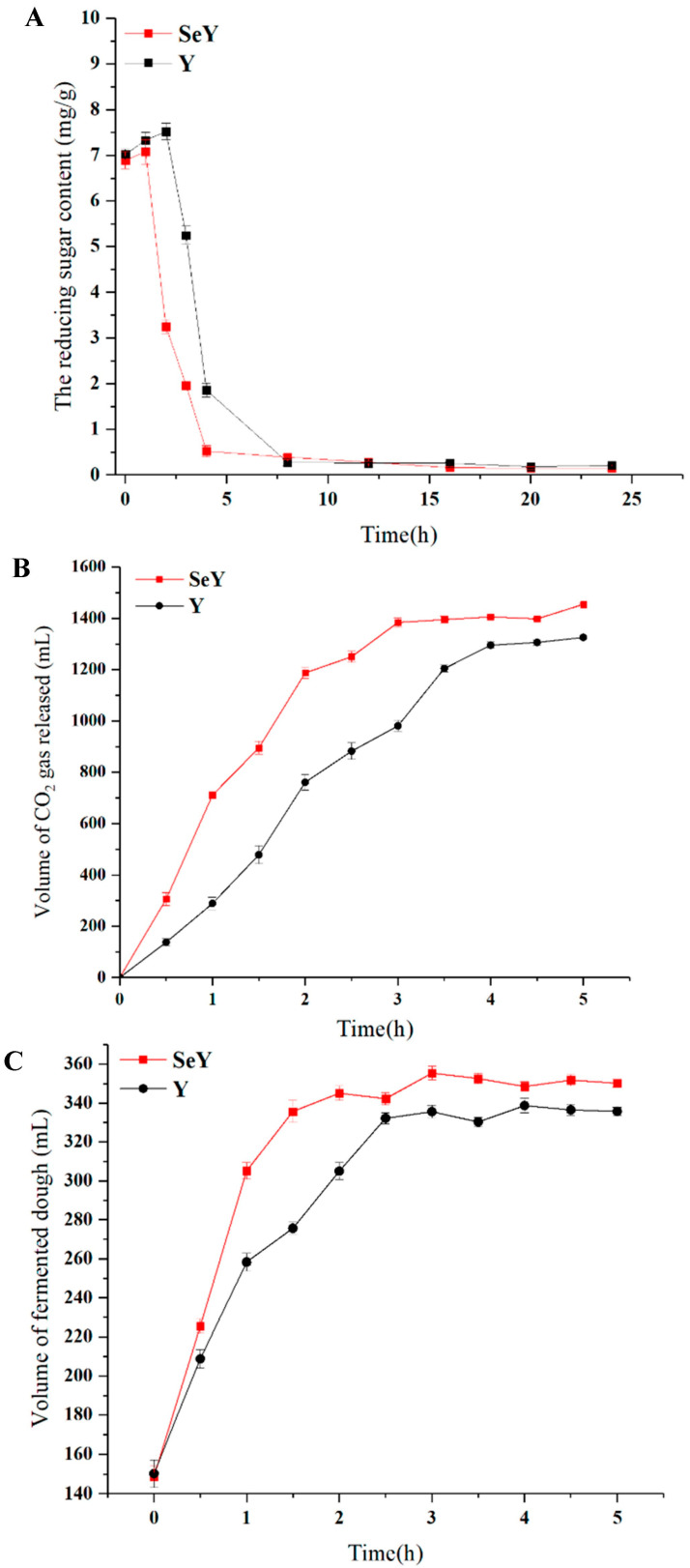
Changes in reducing sugar content (**A**), CO_2_ gas-release volume (**B**), and dough fermentation volume (**C**) for Se-enriched yeast and original yeast strains during dough fermentation.

**Figure 2 foods-12-02343-f002:**
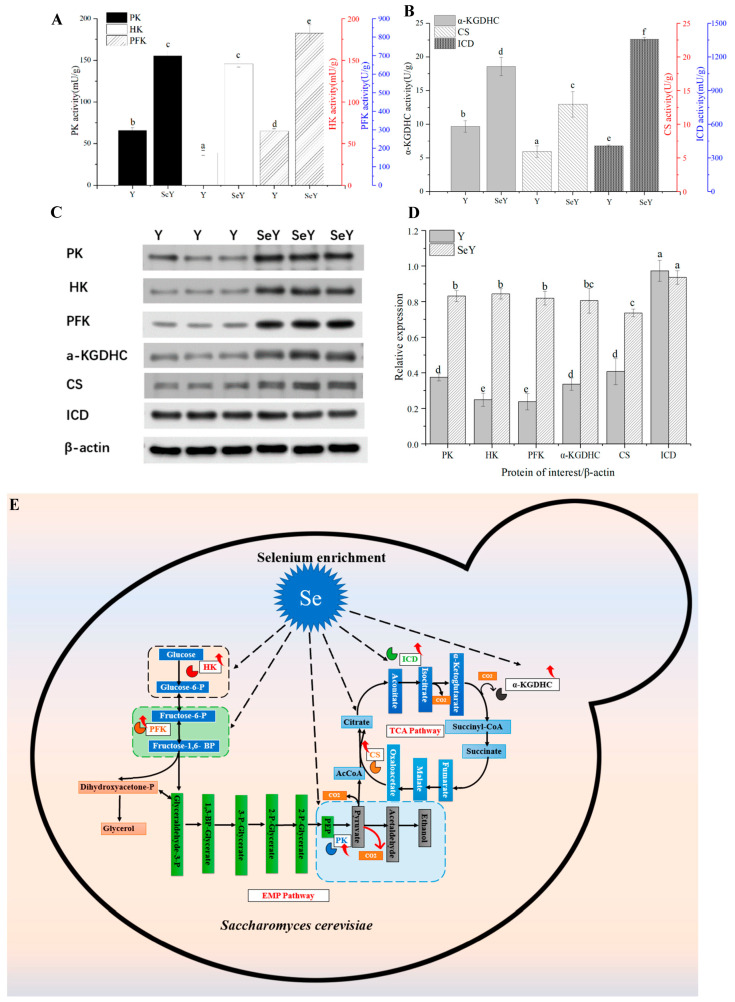
Comparison of specific activity (**A**,**B**), protein expression (**C**), and relative expression (**D**) of six enzymes (PK, HK, PFK, a-KGDHC, CS, ICD) in Se-enriched yeast and the original yeast strain, and possible pathways of glycometabolism (**E**) in *Saccharomyces cerevisiae* regulated by selenium enrichment. The data are presented as means standard deviations (*n* = 3). In multi-range analysis, different lowercase letters in columns indicate significant differences between groups (*p* < 0.05).

**Figure 3 foods-12-02343-f003:**
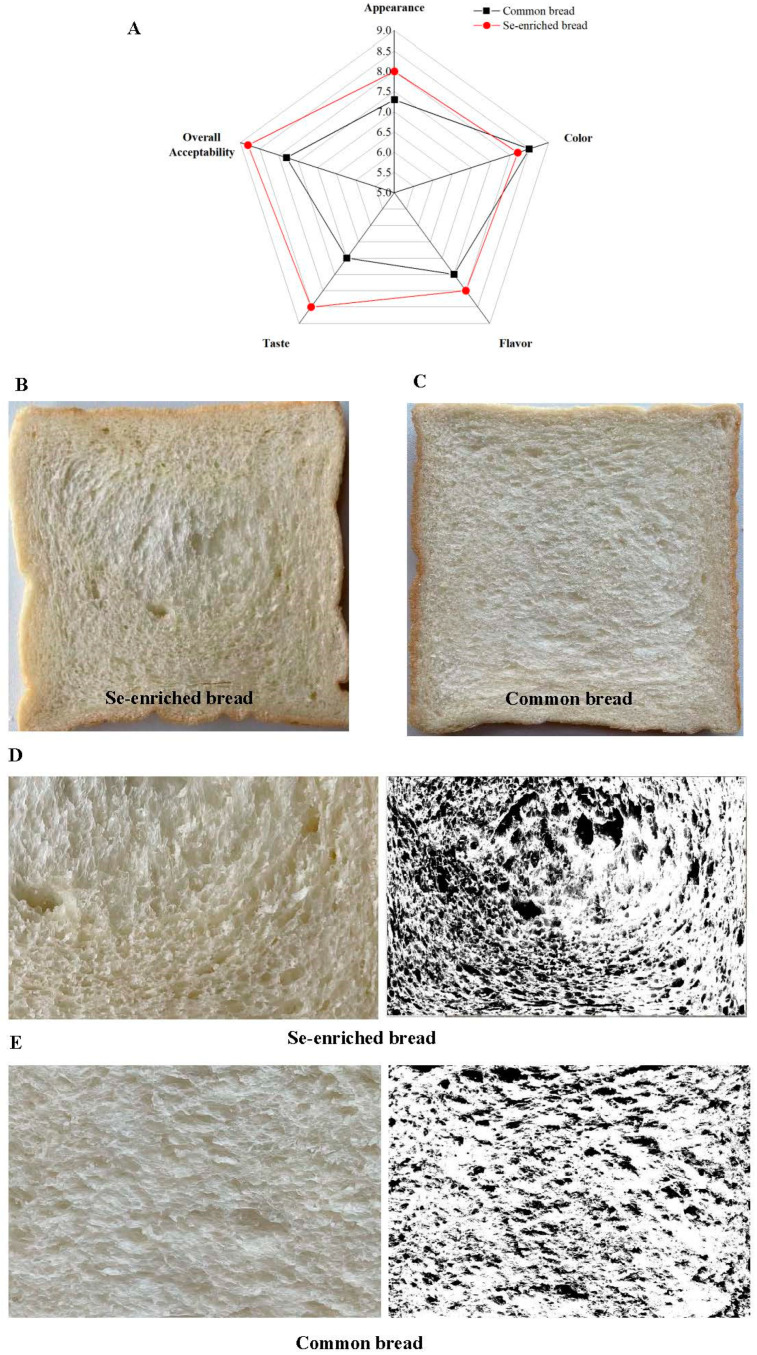
Sensory scores radar chart (**A**) for Se-enriched bread (Se-enriched yeast) and common bread (original yeast strain) evaluated with untrained panelists and a hedonic scale of 1–9, where 1 = dislike extremely; 5 = neither like nor dislike and 9 = like extremely. The apparent structures of the Se-enriched bread (**B**) and common bread (**C**) by HD camera, and the magnification of HD camera is 1×. The visual field of 2 cm × 3 cm in the Se-enriched bread (**D**) and common bread (**E**) center and were processed using ImageJ software.

**Table 1 foods-12-02343-t001:** Rheological fermentation characteristics of Se-enriched yeast during dough fermentation.

Rheological Fermentation Characteristics	Se-Enriched Yeast	Original Yeast Strain
H_m_ (mm)	44.90 ± 0.35 *	39.86 ± 0.28
H′_m_ (mm)	97.48 ± 2.67 *	91.79 ± 1.85
T_1_ (min)	115.75 ± 1.45 *	137.25 ± 2.25
T_x_ (min)	79.50 ± 0.60 *	85.50 ± 0.35
V_gas_ (mL)	1687.53 ± 53.75 *	1588.64 ± 25.86
R (%)	87.15 ± 0.22 *	83.96 ± 0.25

Remarks: H_m_ is the maximum swelling height of dough; H′_m_ is the maximum height of the gas release curve; T_1_ is the time to reach the maximum height of the gas release curve; T_x_ is the time when holes appear in the dough (the time when the dough starts to leak CO_2_); V_gas_ is the total volume of released gas; R is the gas retention in the dough. The asterisk (*) indicates that the experimental group (Se-enriched yeast) had a significant difference compared with the control group (original yeast strain), *p* < 0.05.

**Table 2 foods-12-02343-t002:** Se content, speciation, and bioaccessibility in Se-enriched bread processed by using Se-enriched yeast.

	Total Se Content (μg/g)	Inorganic Se Content (μg/g)	Organic Se Content (μg/g)	SeMet Content (μg/g)	SeMet Percentage (%)	Se Bioaccessibility	
						Total Se Bioaccessibility (%)	SeMet Bioaccessibility (%)
Common bread	0.14 ± 0.03	ND	0.14 ± 0.03	ND	ND	35.71	ND
Se-enriched bread	11.29 ± 0.21 *	ND	11.29 ± 0.21 *	9.24 ± 0.17 *	81.84 ± 3.03 *	94.15 *	97.51

The asterisk (*) indicates that the experimental group (Se-enriched bread) had a significant difference compared with the control group (Common bread), *p* < 0.05. The “ND” means not found.

**Table 3 foods-12-02343-t003:** Effect of Se-enriched yeast fermentation on the stomatal morphology and textural properties of bread.

Properties	Se-Enriched Bread	Common Bread
Cell density (cell/cm^2^)	144.17 ± 1.85 *	106.74 ± 1.42
Average stomatal area (mm^2^)	0.32 ± 0.0012 *	0.24 ± 0.0021
Area fraction (%)	32.10 ± 0.35 *	23.29 ± 0.25
Hardness (g)	113.91 ± 3.33 *	192.67 ± 3.49
Elasticity (mm)	1.36 ± 0.020 *	0.93 ± 0.031
Cohesiveness (mJ)	0.90 ± 0.0056 *	0.79 ± 0.0044
Chewiness (mJ)	102.68 ± 4.32 *	114.75 ± 2.19
Resilience	0.55 ± 0.063 *	0.56 ± 0.021

The asterisk (*) indicates that the experimental group (Se-enriched bread) had a significant difference compared with the control group (common bread), *p* < 0.05.

## Data Availability

Data is contained within the article or Appendix A.

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
