# Peer review of "Effects of Selenium Enrichment on Dough Fermentation Characteristics of Baker’s Yeast"

_foods, 2023, doi:10.3390/foods12122343_

Round 1
Reviewer 1 Report
The text must be thoroughly checked. There are some typos. words are adjacent to each other. these need to be corrected. The introduction part should be prepared in the form of disposition according to the importance of the subject. Similar sentences should be gathered in one paragraph. Results should be highlighted. There has not been enough discussion in the literature.
-Title: The title is too complicated. should be simplified
-In abstract: This sentences should be transferred to Introduction: “Selenium (Se)-enriched yeast was often used as Se-supplements in food-field. However, 15 few studies have focused on effects of Se enrichment on dough fermentation characteristics of yeast, 16 and the application of se-enriched yeast as starter in baked-foods”.
-Did Selenium cause the increase in carbon dioxide in Hanur or was it due to the optimum environmental conditions for the sugars in the flour to be fermented by yeasts? What is the mechanism of selenium overconsumption of sugar by yeasts?
-The presence of protein in the environment does not prevent the formation of carbon dioxide. Why do enzymes suppress protein?
- Sub title 2.4: It is time consuming for the yeasts to act directly on the garnul sugar. First, the granulated sugar must be inverted so that the invert sugars become active and are more easily consumed by the yeasts and carbon dioxide is released. Why was such a path not followed?
- Why wasn't a control group created?
The text must be thoroughly checked. There are some typos. words are adjacent to each other. these need to be corrected. The introduction part should be prepared in the form of disposition according to the importance of the subject. Similar sentences should be gathered in one paragraph. Results should be highlighted. There has not been enough discussion in the literature.
-Title: The title is too complicated. should be simplified
-In abstract: This sentences should be transferred to Introduction: “Selenium (Se)-enriched yeast was often used as Se-supplements in food-field. However, 15 few studies have focused on effects of Se enrichment on dough fermentation characteristics of yeast, 16 and the application of se-enriched yeast as starter in baked-foods”.
-Did Selenium cause the increase in carbon dioxide in Hanur or was it due to the optimum environmental conditions for the sugars in the flour to be fermented by yeasts? What is the mechanism of selenium overconsumption of sugar by yeasts?
-The presence of protein in the environment does not prevent the formation of carbon dioxide. Why do enzymes suppress protein?
- Sub title 2.4: It is time consuming for the yeasts to act directly on the garnul sugar. First, the granulated sugar must be inverted so that the invert sugars become active and are more easily consumed by the yeasts and carbon dioxide is released. Why was such a path not followed?
- Why wasn't a control group created?
Author Response
Dear editor and reviewers,
Thank you for your efforts on our manuscript. We also much appreciate the meticulous review provided by reviewers. We have worked on your comments carefully and have made corrections and hope this version could meet with your approval. Revised parts are highlighted in red in the paper, with main corrections and the response s for the reviewer’s comments are as follows:
Reviewer #1:
- The text must be thoroughly checked. There are some typos. words are adjacent to each other. these need to be corrected. The introduction part should be prepared in the form of disposition according to the importance of the subject. Similar sentences should be gathered in one paragraph. Results should be highlighted. There has not been enough discussion in the literature.
Response: Thank you for your careful examination of our manuscript. According to your suggestion, we invited a partner who is good at English to check the whole article with us, and corrected the problems including typos, adjacent words, language repetition and so on. At the same time, according to the focus of the article, the introduction has been revised and some research status and background on the effects of selenium on glucose metabolism have been added. In addition, sentences with similar results were partially deleted and summarized. Finally, the presentation framework of the results and discussion section of the article was readjusted, and more discussions were added to highlight the key points. the changed part is highlighted in red letters in the text.
- -Title: The title is too complicated. should be simplified
Response: Thank you for your suggestion. The title has been simplified and revised in line 2-3.
- -In abstract: This sentences should be transferred to Introduction: “Selenium (Se)-enriched yeast was often used as Se-supplements in food-field. However, 15 few studies have focused on effects of Se enrichment on dough fermentation characteristics of yeast, 16 and the application of se-enriched yeast as starter in baked-foods”.
Response: Thank you for your comment. The sentences has been transferred to Introduction in line 45-46 and 59-60.
- -Did Selenium cause the increase in carbon dioxide in Hanur or was it due to the optimum environmental conditions for the sugars in the flour to be fermented by yeasts? What is the mechanism of selenium overconsumption of sugar by yeasts?
Response: Thank you for your comment. The responses are as follows:
(1) About “Did Selenium cause the increase in carbon dioxide in Hanur or was it due to the optimum environmental conditions for the sugars in the flour to be fermented by yeasts?”
In this study, original yeast and Se-enriched yeast were used for dough fermentation respectively, both of which were used in the same environment, and other conditions and parameters within the fermentation system were the same. The only difference is that one is the yeast after selenium enrichment (Se-enriched yeast), and the other is the original yeast before selenium enrichment (original yeast). Therefore, the difference generated during the dough fermentation process should be caused by selenium.
(2) About “What is the mechanism of selenium overconsumption of sugar by yeasts?”
Inorganic selenium is often toxic. In order to alleviate this toxicity, yeast converts this selenium into organic selenium, which is selenoamino acids, including selenomethionine and selenocysteine. This will greatly promote the synthesis of methionine or cysteine in yeast, which will lead to the inclination of carbon metabolism in yeast towards the synthesis of amino acids. Therefore, in order to maintain intracellular carbon balance, yeast needs to consume more sugar to maintain its energy demand.
Specifically, for the synthesis of amino acids, the raw materials required by its synthesis pathway are the substances produced by some carbon metabolism pathways of yeast. Therefore, the synthesis of methionine and cysteine will consume products of the tricarboxylic acid cycle or other carbon metabolism cycles. Then, some substances in the carbon cycle will be consumed in large quantities, which will lead to the disorder of sugar metabolism in yeast. Finally, in order to balance it, yeast will consume more carbohydrates (sugars) to supplement the consumption of these metabolites.
- -The presence of protein in the environment does not prevent the formation of carbon dioxide. Why do enzymes suppress protein?
Response: Thank you for your comment. Due to our limited understanding, we could not fully understand the reviewer's meaning. We would appreciate it if you could raise this question again and point out which part of the manuscript is convenient for us to understand.
- - Sub title 2.4: It is time consuming for the yeasts to act directly on the garnul sugar. First, the granulated sugar must be inverted so that the invert sugars become active and are more easily consumed by the yeasts and carbon dioxide is released. Why was such a path not followed?
Response: Thank you for your comment. As reviewer have pointed out, the action of yeast on the granulated sugar is time-consuming. If sugar inverted from the granulated sugar is directly used, it is more easily to consumed by yeast and release carbon dioxide. However, in the actual production process of bread, the main use is the granulated sugar. In this study, we mainly wanted to explore the actual application of Se-enriched yeast in bread making, rather than exploring how yeast can more efficiently ferment dough. Therefore, we directly used the granulated sugar, just to get closer to the actual production.
- - Why wasn't a control group created?
Response: Thank you for your comment. In this study, the original strain of se-enriched yeast was used as the control.
We have tried our best to improve the manuscript and made some changes in the manuscript. We appreciate for Reviewers’ warm work earnestly, and hope the corrections meet with your approval. Those comments are all valuable and constructive to help us revise the paper and improve the study, as well as important guidance for our researches. Finally, the authors would like to thank you very much again for your comments and suggestions.
Yours sincerely,
Hui Wu

Reviewer 2 Report
Dear Authors,
The work is really very interesting, however some minor modifications are necessary.
Keywords: I would recommend not putting the same keywords that are already in the title
Abstract
Line 18: Delete the term S.cerevisiae.
Line 43: Insert the reference for this sentence.
Line 53: Delete the comma.
Line 56: Insert the reference for this sentence.
Line 78: Insert the reference for this sentence.
Line 83: Insert full name and abbreviations in brackets.
Line 93: Delete “and”.
Line 113: Delete the comma after the term “protein”.
Line 114-115: Can't you move the period in brackets in the 'Results/Discussion' section?
Line 125: Write the period better.
Line 136-137: Explain the procedure better. Why write about rapeseed oil?
Line 152: Specify full names of enzymes when first entered in the text.
Line 212: Add the number of replicate.
Materials and methods
Line 131: What is the range of the calibration curve and the R2? Was only glucose evaluated as reduced sugar?
Line 180: Delete the comma.
Line 249: Delete “and” after the term “gluten”.
Line 430-432: To verify the font.
Table 2 Adjust the formatting of the table and indicate below the table what “ND” means.
Line 528-535: With reference to this period and to the cited reference, was it not possible to evaluate the aromatic profile of the bread instrumentally (for example with GC-MS)?
References
Follow the guidelines of journal.
Author Response
Dear editor and reviewers,
Thank you for your efforts on our manuscript. We also much appreciate the meticulous review provided by reviewers. We have worked on your comments carefully and have made corrections and hope this version could meet with your approval. Revised parts are highlighted in red in the paper, with main corrections and the response s for the reviewer’s comments are as follows:
Reviewer #2:
Dear Authors,
The work is really very interesting, however some minor modifications are necessary.
Thank you for providing a detailed review of our manuscript. The response is as follows:
- Keywords: I would recommend not putting the same keywords that are already in the title
Response: Thank you for your suggestion. The keywords has been revised in line 27.
Abstract
- Line 18: Delete the term S.cerevisiae.
Response: Thank you for your comment. According to the journal Guide, it is required to mark the abbreviation when it first appears in the manuscript, so there is no need to remove the abbreviation here.
- Line 43: Insert the reference for this sentence.
Response: Thank you for your comment. The reference has been added in line 40.
4.Line 53: Delete the comma.
Response: Thank you for your comment. The comma has been deleted in line 49.
- Line 56: Insert the reference for this sentence.
Response: Thank you for your comment. The reference has been added in line 52.
- Line 78: Insert the reference for this sentence.
Response: Thank you for your comment. The reference has been added in line 103.
- Line 83: Insert full name and abbreviations in brackets.
Response: Thank you for your comment. The abbreviation was already marked with its full name when it first appeared in the manuscript (line 20-22), so there is no necessary to insert the full name again here.
- Line 93: Delete “and”.
Response: Thank you for your comment. The “and” has been deleted in line 117.
- Line 113: Delete the comma after the term “protein”.
Response: Thank you for your comment. The comma after the term “protein” has been deleted in line 137.
- Line 114-115: Can't you move the period in brackets in the 'Results/Discussion' section?
Response: Thank you for your comment. We have moved the period in brackets to 'Results and discussion' section in line 247-250.
- Line 125: Write the period better.
Response: Thank you for your comment. The period has been rewritten in line 146-151.
- Line 136-137: Explain the procedure better. Why write about rapeseed oil?
Response: Thank you for your comment. Rapeseed oil is a medium used by a food volume analyzer to measure the volume of dough. The dough was placed in the sample room of a food volume analyzer, and the volume occupied by the dough will squeeze rapeseed oil in sample room into a glass graduated tube. Then, record the volume of the dough (mL) ac-cording to the scale. We have provided a more detailed description of the procedure in line 160-163.
- Line 152: Specify full names of enzymes when first entered in the text.
Response: Thank you for your comment. The abbreviation was already marked with its full name when it first appeared in the manuscript (line 20-22).
- Line 212: Add the number of replicate.
Response: Thank you for your comment. The number of replicate has been added in line 238.
Materials and methods
- Line 131: What is the range of the calibration curve and the R2? Was only glucose evaluated as reduced sugar?
Response: Thank you for your comment. The range of the calibration curve and the R2 are shown in Table 1. Since the data from Table 1 is not the focus of this study, it was not included in the manuscript.
The fructose, glucose and maltose are evaluated as reduced sugar. Due to omissions in our previous expression in this part, we have made new modifications in line 154-155.
Table 1 Standard curve for quantifying the reduced sugar
|
Fructose |
Glucose |
Maltose |
||||||
|
Concentration (mg/mL) |
Peak area |
Time (min) |
Concentration (mg/mL) |
Peak area |
Time (min) |
Concentration (mg/mL) |
Peak area |
Time (min) |
|
0.20 |
1491529 |
8.836 |
0.2 |
726456 |
12.187 |
0.20 |
1314527 |
16.751 |
|
0.40 |
3406321 |
8.898 |
0.4 |
1870734 |
12.127 |
0.40 |
3111333 |
16.69 |
|
0.60 |
5322565 |
8.881 |
0.6 |
3220969 |
12.14 |
0.60 |
5065477 |
16.643 |
|
0.80 |
7343848 |
8.856 |
0.8 |
4550809 |
12.102 |
0.80 |
7055860 |
16.60 |
|
1.00 |
9243420 |
8.836 |
1 |
6109069 |
12.1 |
1.00 |
9012172 |
16.537 |
|
y = 9720654.50 x - 470856.10 R2=0.9999 |
y = 6,722,650.50 x - 737982.90 R2=0.9972 |
y = 9669908.50 x - 690071.30 R2=0.9997 |
||||||
Line 180: Delete the comma.
Response: Thank you for your comment. The comma has been deleted in line 206.
- Line 249: Delete “and” after the term “gluten”.
Response: Thank you for your comment. The “and” has been deleted in line 285.
- Line 430-432: To verify the font.
Response: Thank you for your comment. The font format has been changed in line 494-497.
- Table 2 Adjust the formatting of the table and indicate below the table what “ND” means.
Response: Thank you for your comment. We have made formatting changes to Table 2 as required. The meaning of ND has been added to the notes in Table 2.
- Line 528-535: With reference to this period and to the cited reference, was it not possible to evaluate the aromatic profile of the bread instrumentally (for example with GC-MS)?
Response: Thank you for your comment. It is possible to evaluate the aromatic profile of the bread instrumentally through GC-MS or other means. However, in this study, for Se-enriched bread, we mainly want to explore the overall acceptance of consumers. Therefore, we care not only about flavor substances, but also appearance, taste and other aspects, so we choose the method of sensory evaluation for comprehensive evaluation. Morever, the specific composition of flavor substances in Se-enriched bread is not our main work in our research, but this work is very interesting and valuable. It is worth further research and exploration in our future work.
References
- Follow the guidelines of journal
Response: Thank you for your comment. We have thoroughly reviewed and revised the format of the references as required.
We have tried our best to improve the manuscript and made some changes in the manuscript. We appreciate for Reviewers’ warm work earnestly, and hope the corrections meet with your approval. Those comments are all valuable and constructive to help us revise the paper and improve the study, as well as important guidance for our researches. Finally, the authors would like to thank you very much again for your comments and suggestions.
Yours sincerely,
Hui Wu

Round 2
Reviewer 1 Report
All corrections were made by authors